# VATS Pleurectomy Decortication Is a Reasonable Alternative for Higher Risk Patients in the Management of Malignant Pleural Mesothelioma: An Analysis of Short-Term Outcomes

**DOI:** 10.3390/cancers13051068

**Published:** 2021-03-03

**Authors:** Dong-Seok Lee, Andrea Carollo, Naomi Alpert, Emanuela Taioli, Raja Flores

**Affiliations:** Thoracic Surgery Department, Icahn School of Medicine at Mount Sinai, Mount Sinai Health System, New York, NY 10029, USA; Andrea.Carollo@mountsinai.org (A.C.); Naomi.Alpert@mountsinai.org (N.A.); Emanuela.Taioli@mountsinai.org (E.T.); Raja.Flores@mountsinai.org (R.F.)

**Keywords:** malignant mesothelioma, VATS, extrapleural pneumonectomy, pleurectomy decortication

## Abstract

**Simple Summary:**

Malignant pleural mesothelioma (MPM) is an aggressive malignancy that drastically affects a patient’s quality of life. Surgery typically entails radical resection with or without the removal of the underlying lung. In an era where minimally invasive surgery is sought after, MPM remains an anomaly. The purpose of this study is to assess the feasibility of minimally invasive surgery as an alternative to more radical surgery in MPM. We examined short-term outcomes between the radical approaches and minimally invasive surgery and minimally invasive surgery had improved outcomes. Minimally invasive surgery can be considered in patients with MPM.

**Abstract:**

Surgery is a mainstay of treatment allowing for debulking of tumor and expansion of the lung for improvement in median survival and quality of life for patients with malignant pleural mesothelioma (MPM). Although optimal surgical technique remains open for debate—extrapleural pneumonectomy (EPP) vs. pleurectomy/decortication (P/D)—minimally invasive surgery (VATS-P/D) remains underutilized in the management of MPM. We examined whether VATS-P/D is a feasible alternative to EPP and P/D. We evaluated the New York Statewide Planning and Research Cooperative System (SPARCS) from 2007–2017 to assess the short-term complications of EPP vs. P/D, including a subanalysis of open P/D vs. VATS-P/D. There were 331 patients with open surgery; 269 with P/D and 62 with EPP. There were 384 patients with P/D; 269 were open and 115 VATS. Rates of any complication were similar between EPP and P/D patients, but EPP had significantly higher rates of cardiovascular complications. After adjusting for confounders, those with a VATS approach were less likely to have any complication, compared to an open approach and significantly less likely to have a pulmonary complication. VATS-P/D remains a viable alternative to radical surgery in MPM patients allowing for improved short-term outcomes.

## 1. Introduction

Malignant pleural mesothelioma (MPM) is a rare but aggressive cancer with an overall poor prognosis. Treatment frequently involves multimodal therapy, of which surgical resection remains an essential component, significantly improving median survival compared to patients who do not undergo surgery [1]. However, there remains debate about the optimal surgical technique. Extrapleural pneumonectomy (EPP) theoretically offers the better chance at complete resection and was considered the standard. However, lung-sparing pleurectomy/decortication (P/D) has become more common, as research has indicated decreased perioperative morbidity and mortality and similar survival compared to EPP [2,3,4,5,6]. In addition, quality of life appears better as physical and social function and global health measures are better at 12 months with P/D over EPP [7,8].

Despite the increasing utilization of minimally invasive techniques in many oncologic surgical procedures, MPM-directed surgeries have historically been performed as open procedures. Although minimally invasive lung surgery has improved short-term outcomes with equivalent long-term survival compared to open surgery [9,10], its use in MPM is more challenging. Video-assisted thoracoscopic surgery (VATS) has been primarily focused on diagnosis or palliation of symptoms. Although there is extensive literature comparing outcomes of EPP to P/D, there is a paucity of data examining outcomes of minimally invasive surgery for MPM. Our group had previously reported improved short-term outcomes for patients with P/D compared to EPP using New York State hospital discharge data [3]. The aims of this study were to utilize the same large database to provide updated results of our prior study, with an added focus on comparing a minimally invasive approach to open surgery.

## 2. Materials and Methods

### 2.1. Data Source and Sample Selection

This analysis used the New York Statewide Planning and Research Cooperative System (SPARCS) from 2007–2017. SPARCS includes all hospital discharges in the state, and has information on patient demographics, diagnoses, procedures, admission and discharge type. This research was approved by the Mount Sinai Institutional Review Board (IRB# 18-00947, FWA #00005656).

There were 4,959,270 patients at least 50 years old, with a patient identifier who had an inpatient discharge between 1 January 2007 and 31 December 2017. Those with an admission accompanied by a diagnosis of pleural mesothelioma (*n* = 2169) and who had either EPP or P/D (See Appendix A for ICD-9 and ICD-10 diagnosis and procedures codes) were included (*n* = 589) for analysis. For patients with multiple mesothelioma-related surgeries, the first surgery was chosen. Patients where the surgical approach (open or minimally invasive) was unknown were excluded, as were the few who were coded as having minimally invasive EPP (n_excl_ = 143). The initial analysis was limited to patients with an open EPP or P/D surgery (*n* = 331), while a secondary analysis compared surgical approach among those with P/D (*n* = 384) (Figure 1).

### 2.2. Predictors and Outcomes

The primary predictors of interest were the type of surgery and surgical approach. Outcomes of interest were short-term complications after surgery. In-hospital complications were defined based on diagnosis codes that were not present at the time of admission (Appendix A), and were categorized as cardiovascular, pulmonary, infectious or intraoperative complications. Patient comorbidities were defined using the algorithm described by Elixhauser, et al. [11], and a count of non-cancer-related comorbidities was created. Other covariates of interest included age, gender, race (Non-Hispanic White (NHW) vs. Hispanic or Non-White), primary insurance payer (government vs. non-government), type of admission to the hospital (urgent/emergency vs. elective), and the year of surgery.

### 2.3. Statistical Analysis

Patients were compared across surgical type on all variables, using t-tests for continuous variables, and χ_2_-tests for categorical variables. Univariate and multivariable logistic regressions were used to model the independent associations between covariates and type of surgery, using Odds Ratios (ORs) and 95% Confidence Intervals (CI). Multivariable logistic regression models were also used to assess the association of surgical type with having complications (any, cardiovascular, or pulmonary), adjusting for possible confounders. Supraventricular arrhythmia was examined individually as a subset of cardiovascular complications. As there were a very small number of infectious and intraoperative complications, these were individually assessed only at the univariate level. Multivariable models were adjusted for age, gender, race/ethnicity, admission type, insurance, number of comorbidities, and year of surgery, to account for changes over time. Outcomes were also assessed using an optimal propensity matching analysis, with a maximum difference of 0.01, matching on all variables.

Analyses were repeated on the subset of patients with P/D, in order to compare outcomes in patients with minimally invasive and open approaches. All analyses were conducted using SAS software, v 9.4 (SAS Institute, Cary, NC, USA).

## 3. Results

### 3.1. Extrapleural Pneumonectomy vs. Pleurectomy Decortication

There were 331 patients with open surgery; 269 (81.3%) with P/D and 62 (18.7%) with EPP. EPP patients were significantly younger (mean age: 64.6 vs. 69.1 years, *p* < 0.0001), more likely to have non-government insurance coverage (61.3% vs. 44.6%, *p* = 0.0217), and had fewer comorbidities (29.0% vs. 55.4% with ≥2 comorbidities; *p* = 0.0002). EPP patients also more frequently had elective admissions (*p* = 0.0552) (Table 1).

After adjustment, those with EPP were significantly younger (ORadj: 0.91, 95% CI: 0.86–0.96) and significantly less likely to have an urgent or emergency surgery (ORadj: 0.21, 95% CI: 0.05–0.97). There was no significant difference in gender, race/ethnicity, type of insurance, or number of comorbidities (Table 2).

At the univariate level, rates of any complication were similar between EPP and P/D patients (43.5% for EPP vs. 42.0% for P/D; *p* = 0.8248), but EPP had significantly higher rates of cardiovascular complications (32.3% vs. 13.4%; *p* = 0.0004) supraventricular arrhythmia (27.4% vs. 10.0%; *p* = 0.0003), and lower rates of pulmonary complications (21.0% vs. 34.2%; *p* = 0.0439) (Table 1).

In the multivariable analysis, those with EPP were significantly more likely to have any complication (ORadj: 2.12, 95% CI: 1.08–4.18), as well as have cardiovascular complications (ORadj: 5.00, 95% CI: 2.23–11.24), and supraventricular arrhythmia specifically (ORadj: 6.63, 95% CI: 2.64–16.64). There was no significant difference in the odds of a pulmonary complication (Table 3).

After propensity matching, there were 50 EPP and 50 P/D patients, who were well matched on all covariates (range of *p*-values: 0.5637 to 1). Although not statistically significant, patients with EPP continued to have more cardiovascular complications in general (OR: 2.60, 95% CI: 0.93–7.29), and specifically supraventricular arrhythmia (OR: 2.75, 95% CI: 0.88–8.64) (Table 3).

### 3.2. Minimally Invasive vs. Open P/D

There were 384 patients with P/D; 269 (70.1%) with an open surgical approach, and 115 (29.9%) with a minimally invasive approach. Patients with a minimally invasive surgical approach were significantly older (mean age: 71.8 vs. 69.1 years; *p* = 0.0132) and more likely to have an urgent/emergency admission (47.0% vs. 11.2%; *p* < 0.0001). They were also less often NHW (*p* = 0.0524) (Table 4).

After adjustment, those with a minimally invasive approach remained significantly older (ORadj: 1.05, 95% CI: 1.01–1.08) and more likely to have an urgent/emergency admission (ORadj: 7.18, 95% CI: 4.07–12.64), compared to those with an open approach (Table 5).

After adjusting for confounders, those with a minimally invasive approach were less likely to have any complication, compared to those with an open approach (ORadj: 0.58, 95% CI: 0.34–1.01) and significantly less likely to have a pulmonary complication (ORadj: 0.55, 95% CI: 0.31–0.99) (Table 6).

The propensity-matched analysis was well balanced on all covariates (range of *p*-values: 0.3980–1) with 75 patients per group. Although not significant, results were similar, with minimally invasive surgery having lower risk of overall complications (OR: 0.70, 95% CI: 0.37–1.32) and pulmonary complications (OR: 0.65, 95% CI: 0.30–1.38) (Table 6).

## 4. Discussion

Our study utilized the New York SPARCS database in order to compare perioperative morbidity with EPP, P/D, and VATS-P/D for MPM. Complications examined were cardiovascular, pulmonary, infectious, and intraoperative complications. The majority of complications were either cardiovascular or pulmonary. Perioperative mortality was not included in the present analysis due to limited observations. Generally, the more radical resection was associated with younger age, elective procedure, and increased incidence of complications. EPP patients were more likely to have cardiovascular complications, primarily supraventricular arrhythmias, than P/D patients on multivariable analysis and propensity matching. On the other hand, cardiovascular complications were similar in open and minimally invasive P/D patients but open patients were more prone to pulmonary complications on multivariable analysis and propensity matching.

The goal of oncologic surgery with curative intent is removal of all macroscopic and, if possible, microscopic disease. This is challenging in MPM as it is an insidious diffuse disease throughout the pleura and often requires radical resection. Therefore, the mainstay of surgical treatment for MPM includes extrapleural pneumonectomy and pleurectomy/decortication. A number of studies have been performed showing that EPP and P/D confer similar overall survival but that the short-term mortality and morbidity associated with EPP is greater than P/D [2,3,4,5,6]. Less radical and more minimally invasive surgery has primarily been limited to diagnostic biopsy or symptom management with talc pleurodesis or indwelling pleural catheters. VATS- P/D has not achieved widespread use in the management of MPM, as it is primarily considered to be a palliative surgical option [12] as opposed to a potentially curative one. The goal of VATS- P/D is the debulking of enough pleural disease and decortication of the underlying trapped lung in order to obliterate the pleural space to allow pleural apposition.

The only randomized control trial to date, MesoVATS, compared VATS partial pleurectomy (VATS-PP) to talc pleurodesis [13]. The primary endpoint was overall survival at 12 months and no significant difference was noted between the two groups. Although VATS-PP had a non-significant trend towards increased morbidity, the authors noted a 70% resolution of pleural effusion with VATS-PP compared to 77% resolution with talc pleurodesis but significantly improved quality of life scores at 6 and 12 months for the VATS-PP group. A follow-up study, currently in progress, aims to address VATS-PP against the use of indwelling pleural catheters for patients with MPM and trapped lung [14].

In addition to providing a palliative benefit, VATS-P/D appears to confer a survival benefit as cytoreduction and post-resection tumor volume may play a role in long-term outcomes [15,16]. It is unclear how it compares with more radical surgery. A previously published single institutional study looking at VATS P/D showed a modest non-significant improvement in survival with VATS versus EPP (14 months vs. 11.5 months). They also noted symptomatic improvement in the majority of patients and statistically significant advantage in 30-day mortality versus EPP [17].

Our study is the first to utilize a large population-based database in order to assess short-term outcomes in EPP, P/D, and minimally invasive P/D. However, it is not without its limitations. Despite the extensive size of the dataset, MPM remains an uncommon disease such that it accounts for a very small percentage of admissions, and thus, numbers remain relatively small. There may be selection bias in regards to surgical technique due to both surgeon preference and elective versus emergent presentation. Confounders that are unable to be addressed include information that could not be ascertained from the database, such as tumor grade, oncologic stage, long-term outcomes, surgeon experience, and potential use of induction therapy. However, this analysis includes a greater number of patients than would be available from a single-center study.

In confirmation of our previous analysis, P/D was associated with improved short-term outcomes compared to EPP and likely explains the shift from equivalent amounts of EPP and P/D performed (46.6% EPP, 53.4% P/D) from 1995–2012 [3] to predominantly P/D (81.3% P/D, 18.7% EPP) performed for the treatment of MPM from 2007–2017. Despite the increasing age of patients with less radical surgery, VATS P/D patients exhibited improved short-term outcomes, when controlling for this difference. Further investigation in regards to long-term survival with VATS P/D in comparison to EPP and P/D is needed.

## 5. Conclusions

Malignant pleural mesothelioma remains a challenging cancer to treat. Surgical options range from the more radical curative techniques such as EPP and P/D to the less invasive palliative VATS P/D. Patients who undergo VATS P/D have better short-term outcomes compared to those who undergo curative attempts at surgery. Therefore, VATS P/D should be considered in the armamentarium of treatment for MPM, especially in older and frailer patients who may not tolerate more radical surgery.

## Figures and Tables

**Figure 1 cancers-13-01068-f001:**
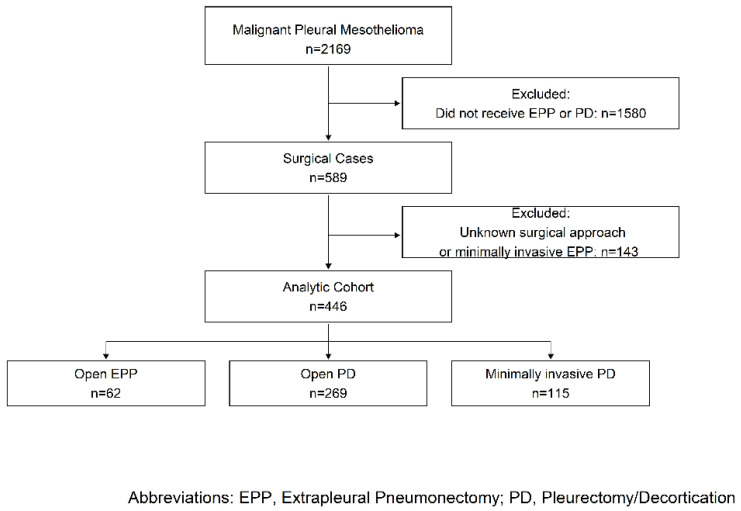
Patient Selection.

**Table 1 cancers-13-01068-t001:** Demographics of the sample, according to surgery type.

Variable	P/D (*n* = 269)	EPP (*n* = 62)	*p*-Value
*Patient and Admission Characteristics*	N (%)	N (%)	
Mean Age, years (SE)	69.1 (0.5)	64.6 (0.8)	<0.0001
Gender			0.9314
Male	201 (74.7)	46 (74.2)	
Female	68 (25.3)	16 (25.8)	
Race			0.1726
NHW	214 (79.6)	≥11 *	
Hispanic or Non-White	55 (20.4)	<11 *	
Primary Insurance Payer			0.0217
Non-Government	120 (45.1)	38 (61.3)	
Government	146 (54.9)	24 (38.7)	
Type of Admission			0.0552
Elective	237 (88.8)	≥11 *	
Urgent/Emergency	30 (11.2)	<11 *	
Number of Comorbidities			0.0002
0–1	120 (44.6)	44 (71.0)	
≥2	149 (55.4)	18 (29.0)	
*Complications*			
Cardiovascular	36 (13.4)	20 (32.3)	0.0004
Pulmonary	92 (34.2)	13 (21.0)	0.0439
Infection	13 (4.8)	<11 *	0.1395
Bleeding	<11 *	<11 *	0.4381
Supraventricular arrhythmia			0.0003
No	242 (90.0)	45 (72.6)	
Yes	27 (10.0)	17 (27.4)	
Any Complication			0.8248
No	156 (58.0)	35 (56.5)	
Yes	113 (42.0)	27 (43.5)	

Abbreviations: P/D, Pleurectomy Decortication; EPP, Extrapleural pneumonectomy. * Exact cell sizes masked to protect against identification of patients.

**Table 2 cancers-13-01068-t002:** Independent Factors Associated with Receipt of EPP vs. P/D (*n* = 326).

	EPP vs. P/D	
Variable	OR_adj_ * (95% CI)	*p*-Value
Age (years)	0.91 (0.86–0.96)	0.0011
Gender		
Female vs. Male	0.88 (0.42–1.84)	0.7347
Race/Ethnicity		
Hispanic or Non-White vs. Non-Hispanic White	0.57 (0.22–1.45)	0.2354
Admission Type		
Urgent/Emergency vs. Elective	0.21 (0.05–0.97)	0.0450
Insurance		
Non-Government vs. Government	0.82 (0.37–1.79)	0.6103
Number of Comorbidities		
≥2 vs. 0–1	0.62 (0.32–1.22)	0.1637

* Adjusted for all variables listed and year of surgery.

**Table 3 cancers-13-01068-t003:** Odds of Complications in EPP vs. P/D patients, multivariable and propensity-matched analyses.

	Any Complication (Y vs. N)	Cardiovascular Complication (Y vs. N)	Supraventricular Arrhythmia (Y vs. N)	Pulmonary Complication (Y vs. N)
	OR_adj_ * (95% CI); *p*-Value	OR_adj_ * (95% CI); *p*-Value	OR_adj_ * (95% CI); *p*-Value	OR_adj_ * (95% CI); *p*-Value
*Multivariable Analysis* (*n = 326*)				
EPP vs. P/D	2.12 (1.08–4.18); 0.0302	5.00 (2.23–11.24); <0.0001	6.63 (2.64–16.64); <0.0001	0.89 (0.41–1.91); 0.7619
*Propensity-Matched Analysis* (*n = 100*)				
EPP vs. P/D	1.11 (0.45–2.73); 0.8186	2.60 (0.93–7.29); 0.0694	2.75 (0.88–8.64); 0.0832	0.58 (0.23–1.48); 0.2571

Abbreviations: EPP, extrapleural pneumonectomy; P/D, Pleurectomy Decortication. * Adjusted for/propensity matched on age, gender, race/ethnicity, admission type, insurance, number of comorbidities, and year of surgery. Adjusted models were not conducted for infection or intraoperative complication due to an insufficient number of outcomes.

**Table 4 cancers-13-01068-t004:** Demographics of the sample according to surgical approach among P/D patients.

Variable	Open (*n* = 269)	Minimally Invasive (*n* = 115)	*p*-Value
*Patient and Admission Characteristics*	N (%)	N (%)	
Mean Age, years (SE)	69.1 (0.5)	71.8 (1.0)	0.0132
Gender			0.5773
Male	201 (74.7)	89 (77.4)	
Female	68 (25.3)	26 (22.6)	
Race			0.0524
NHW	214 (79.6)	81 (70.4)	
Hispanic or Non-White	55 (20.4)	34 (29.6)	
Primary Insurance Payer			0.1194
Non-Government	120 (45.1)	42 (36.5)	
Government	146 (54.9)	73 (63.5)	
Type of Admission			<0.0001
Elective	237 (88.8)	60 (52.6)	
Urgent/Emergency	30 (11.2)	54 (47.4)	
Number of Comorbidities			0.7899
0–1	120 (44.6)	53 (46.1)	
≥2	149 (55.4)	62 (53.9)	
*Complications*			
Cardiovascular	36 (13.4)	11 (9.6)	0.2958
Pulmonary	92 (34.2)	31 (27.0)	0.1635
Infection	13 (4.8)	<11 *	0.8369
Bleeding	<11 *	<11 *	0.7296
Any Complication			0.0995
No	156 (58.0)	77 (67.0)	
Yes	113 (42.0)	38 (33.0)	

Abbreviations: P/D, Pleurectomy Decortication * Exact cell sizes masked to protect against identification of patients. Percentages and *p*-values are presented for non-missing values.

**Table 5 cancers-13-01068-t005:** Independent Factors Associated with Receipt of Minimally Invasive vs. Open Surgery (*n* = 378).

	Minimally Invasive vs. Open	
Variable	OR_adj_ * (95% CI)	*p*-Value
Age (years)	1.05 (1.01–1.08)	0.0106
Gender		
Female vs. Male	0.90 (0.49–1.64)	0.7343
Race/Ethnicity		
Hispanic or Non-White vs. Non-Hispanic White	1.37 (0.76–2.49)	0.3008
Admission Type		
Urgent/Emergency vs. Elective	7.18 (4.07–12.64)	<0.0001
Insurance		
Non-Government vs. Government	1.11 (0.61–2.00)	0.7353
Number of Comorbidities		
≥2 vs. 0–1	0.66 (0.40–1.10)	0.1126

* Adjusted for all variables listed and year of surgery.

**Table 6 cancers-13-01068-t006:** Odds of complications in minimally invasive vs. open P/D patients, multivariable and propensity-matched analyses.

	Any Complication (Y vs. N)	Cardiovascular Complication (Y vs. N)	Pulmonary Complication (Y vs. N)
	OR_adj_ * (95% CI); *p*-Value	OR_adj_ * (95% CI); *p*-Value	OR_adj_ * (95% CI); *p*-Value
*Multivariable Analysis* (*n = 378*)			
Minimally Invasive vs. Open	0.58 (0.34–1.01); 0.0524	0.88 (0.40–1.95); 0.7518	0.55 (0.31–0.99); 0.0448
*Propensity-Matched analysis* (*n = 150*)			
Minimally Invasive vs. Open	0.70 (0.37–1.32); 0.2649	1.13 (0.43–2.92); 0.8085	0.65 (0.30–1.38); 0.2606

Abbreviations: P/D, Pleurectomy Decortication. * adjusted for/propensity matched on age, gender, race/ethnicity, admission type, insurance, number of comorbidities, and year of surgery. Adjusted models were not conducted for infection or intraoperative complication, due to an insufficient number of outcomes.

## Data Availability

No new data was generated by the authors of this study. The data used and analyzed during the current study are available from the New York State Department of Health.

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
