# Peer review of "VATS Pleurectomy Decortication Is a Reasonable Alternative for Higher Risk Patients in the Management of Malignant Pleural Mesothelioma: An Analysis of Short-Term Outcomes"

_cancers, 2021, doi:10.3390/cancers13051068_

Round 1
Reviewer 1 Report
This is an interesting study that explors short term outcomes between more extensive procedures vs minimally invasive procedures in the context of aggressive malignancy. The dataset is from the NY state database and examines outcomes via regression analysis as well as propensity matching. The findings are interesting but there are several confounders that ought to be adjusted for including institutional volume of mesothelioma as well surgeon volume for minimally invasive procedures. There Is also no information on tumar grade/stage to allow adequate propensity matching. Further, the relative PSM matches are fewer which limits our ability to draw conclusions. Thus, the findings are at best hypothesis generating given the lack of long term data (was this possible?) as well as data on recurrence rates. Also, how many patients received neoadjuvant treatment
Reviewer 2 Report
In their paper entitled „VATS Pleurectomy Decortication is a Reasonable Alternative for Higher Risk Patients in the Management of Malignant Pleural Mesothelioma: an Analysis of Short Term Outcomes“ the authors evaluate retrospective data from the New York Statewide Planning and Research Cooperative System (SPARCS) from 2007-2017 in order to compare the short term outcome of VATS P/D in comparison to open P/D and EPP for the surgical treatment of malignant pleural mesothelioma. They come to the conclusion that VATS P/D was less likely to carry pulmonary complications as compared to open P/D.
VATS P/D is certainly an underestimated procedure which may have its value in severely compromised patients. Although VATS P/D is probably carried out in most centres under the non-systematic premise of debulking with palliative intent (and even in the present data the supposed curative intent is not proven), its intentional and systematic adoption is not yet established, and therefore the authors are to be congratulated for opening a new therapeutic avenue. Moreover, the present paper is well written and exhaustively referenced.
I have only a few minor suggestions for improvement:
- a flow chart illustrating patient allocation into treatment arms should be shown;
- the p-value of Table 2 needs to be shown in the table itself, not only in the text; the same is valid for Tables 3, 5 & 6.
